# ECMO in Myocardial Infarction-Associated Cardiogenic Shock: Blood Biomarkers as Predictors of Mortality

**DOI:** 10.3390/diagnostics13243683

**Published:** 2023-12-17

**Authors:** Thomas Senoner, Benedikt Treml, Robert Breitkopf, Ulvi Cenk Oezpeker, Nicole Innerhofer, Christine Eckhardt, Aleksandra Radovanovic Spurnic, Sasa Rajsic

**Affiliations:** 1Department of Anesthesiology and Intensive Care Medicine, Medical University Innsbruck, 6020 Innsbruck, Austria; thomas.senoner@i-med.ac.at (T.S.); robert.breitkopf@tirol-kliniken.at (R.B.); nicole.innerhofer@i-med.ac.at (N.I.); christine.eckhardt@i-med.ac.at (C.E.); 2Department of Cardiac Surgery, Medical University Innsbruck, 6020 Innsbruck, Austria; cenk.oezpeker@tirol-kliniken.at; 3Clinic for Infectious and Tropical Diseases, University Clinical Centre of Serbia, 11000 Belgrade, Serbia; spurnic@yahoo.com

**Keywords:** ECMO, extracorporeal membrane oxygenation, va-ECMO, inflammation, procalcitonin, adverse events, complications, mortality

## Abstract

Background: Veno-arterial extracorporeal membrane oxygenation (va-ECMO) can provide circulatory and respiratory support in patients with cardiogenic shock. The main aim of this work was to investigate the association of blood biomarkers with mortality in patients with myocardial infarction needing va-ECMO support. Methods: We retrospectively analyzed electronic medical charts from patients receiving va-ECMO support in the period from 2008 to 2021 at the Medical University Innsbruck, Department of Anesthesiology and Intensive Care Medicine. Results: Of 188 patients, 57% (108/188) survived to discharge, with hemorrhage (46%) and thrombosis (27%) as the most frequent adverse events. Procalcitonin levels were markedly higher in non-survivors compared with survivors during the observation period. The multivariable model identified higher blood levels of procalcitonin (HR 1.01, *p* = 0.002) as a laboratory parameter associated with a higher risk of mortality. Conclusions: In our study population of patients with myocardial infarction-associated cardiogenic shock, deceased patients had increased levels of inflammatory blood biomarkers throughout the whole study period. Increased procalcitonin levels have been associated with a higher risk of mortality. Future studies are needed to show the role of procalcitonin in patients receiving ECMO support.

## 1. Introduction

Extracorporeal membrane oxygenation (ECMO) is used as a temporary measure in cases of severe cardiorespiratory failure, to allow time for the organs to recover, as a bridge to transplantation, or the placement of a permanent assist device. Veno-arterial ECMO (va-ECMO) provides both circulatory and respiratory support and can be applied in patients with severe myocardial infarction (MI) associated with cardiogenic shock refractory to conventional therapy. Despite the fact that the therapy for MI has substantially changed in the past decades, infarct-related cardiogenic shock is still characterized by a high mortality rate. Even though ECMO support has potential advantages in this patient population, mortality rates are still high, ranging from 20% to 70% [1,2].

Based on the data from 583 international Extracorporeal Life Support Organization (ELSO) registry centers, almost 200,000 ECMO runs were registered until the end of 2022, with more than 50,000 runs for pulmonary, almost 48,000 for cardiac support, and 15,000 in the case of extracorporeal cardiopulmonary reanimation. The reported overall survival to hospital discharge or transfer was 54%, being highest for neonatal and pulmonary support [3]. A significant rise in the use of ECMO support has been noted in the past 10 years, with a substantial increase in the number of both ECMO centers and runs during the coronavirus disease 2019 (COVID-19) pandemic [3].

The overall benefit, adverse events, and mortality rate during ECMO support are still the subject of discussion. There have been several attempts to identify predictors for adverse events and unfavorable outcomes during ECMO; however, no consistent results have emerged thus far [4,5,6,7,8,9]. Even though hemorrhage is a well-established predictor of mortality [10,11], unique risk factors for bleeding have not yet been identified [12,13,14,15,16,17]. Recent work showed that patients experiencing thromboembolic complications have lower mortality, questioning the recommended therapeutic anticoagulation during ECMO support [14]. Moreover, the monitoring of anticoagulation and the therapeutic goals for anticoagulation are being questioned, as no validated monitoring tool is yet available, while the evidence on appropriate anticoagulation thresholds is still scarce or contradictory [1,14,16,18,19,20,21]. Due to the ease of obtaining daily measured laboratory parameters, previous studies have focused on comparing different laboratory parameters with the occurrence of complications on a certain day of ECMO support; however, these studies failed to incorporate the entire duration of ECMO support and the days following the termination of this support [12,15,17,22]. Thus, the evidence on the trend of blood biomarkers throughout the va-ECMO support and its potential role in patient outcomes is missing.

In this study, our goal was to investigate the association of commonly used laboratory blood parameters with patient outcomes, especially focusing on the time course during ECMO support. Moreover, we report on the demographic and clinical characteristics of patients suffering from MI and necessitating va-ECMO support.

## 2. Materials and Methods

### 2.1. Patient Population

We analyzed electronic medical records of patients with cardiogenic shock due to MI requiring ECMO support who were treated at the Medical University Innsbruck, Department of Anesthesiology and Intensive Care Medicine. The study period covered 14 years, including all patients receiving ECMO from January 2008 to the end of December 2021. Excluded were patients receiving ECMO support due to indications other than MI-associated cardiogenic shock, necessitating multiple ECMO runs, having support for less than one day, or being younger than 30 years.

We collected detailed information on patient clinical and demographic data. This included data on basic disease, indication for ECMO support, disease severity prior to ECMO initiation, the ICU admission simplified acute physiology III (SAPS III) score, sequential organ failure assessment (SOFA) score, presence of cardiopulmonary reanimation before or during ECMO initiation, ECMO support duration, adverse events (including date of onset, type, and location), use of anticoagulation, laboratory parameters (coagulation status with fibrinogen (mg/dL), platelet count (g/L), antithrombin (%), rotational thromboelastometry (ROTEM), international normalized ratio, erythrocytes (T/L), hematocrit (l/L), leucocyte count (g/L), hemoglobin (g/L), procalcitonin (PCT, µg/L), and C-reactive protein (CRP, mg/dL). Finally, information on the cause and the date of death was obtained.

We collected all laboratory data within 24 h prior to ECMO support initiation (baseline), then daily throughout ECMO support, and on the third and tenth day following ECMO termination. The observation was confined to a maximum of ten days, based on the average ECMO duration. Moreover, more than 95% of analyzed patients required support for less than 14 days.

Myocardial infarction was defined (according to the management of acute coronary syndrome, European Society of Cardiology (ESC) Guidelines) as necrosis of cardiomyocyte in the setting of acute myocardial ischemia. This includes type 1 MI (MI due to atherothrombotic events) and types 2–5 MI (including other causes of myocardial ischemia) [23].

Medical records were independently analyzed by two authors (SR, BT), who extracted clinical and demographic data. This work is prepared according to the strengthening of the reporting of observational studies in epidemiology (STROBE) statement checklist of items (Appendix A) [24].

### 2.2. ECMO Management and Anticoagulation

ECMO support is constantly available in our university hospital. The decision to initiate ECMO was reached by mutual judgment of a cardiac anesthesiologist, an intensive care specialist, and a cardiac surgeon. We used ECMO system consisting of a centrifugal pump with an oxygenator (hollow fiber), a heparin-coated circuit, and venous and arterial cannulas, and temperature regulation was provided by an integrated heat exchanger.

The decision to substitute blood and/or coagulation products was at the discretion of the treating physician as well as based on institutional standard operating procedures. At our institution, the hemoglobin level was maintained above 8 g/dL, and bedside coagulation monitoring was used. We employed an individualized approach directed towards tailoring coagulation management according to the underlying disease and patients’ characteristics.

Anticoagulation protocol was based on the ELSO Anticoagulation Guideline and as per the institutional standard operating procedure [18,25]. We used unfractionated heparin as the first-line anticoagulant (with a target aPTT of 50–70 s). Argatroban was used when the anticoagulation with unfractionated heparin (UFH) was inadequate or in cases of heparin-induced thrombocytopenia type II. In case of severe bleeding, continuous anticoagulation was stopped. Anticoagulation was monitored based on the aPTT, anti-factor Xa, ACT, ROTEM^®^, or argatroban blood concentration, and adaptations were made accordingly.

When signs of improved cardiac function (based on echocardiographic examination) were detected, weaning protocol was implemented by a stepwise extracorporeal blood flow reduction. Following joint clinical judgment, a trial off was initiated by reducing the blood flow to below 30% of total. In the case of futility (due to irreversible heart or lung damage, severe brain damage, or multiple organ dysfunction syndrome), ECMO was terminated promptly. In certain cases, patients were evaluated for potential organ explanation and donation.

### 2.3. Outcomes

The main endpoint of this work was the association of routinely measured blood parameters with in-hospital mortality. Secondary endpoints covered the clinical characteristics as well as the rate and type of unfavorable events during ECMO.

Analyzed adverse events included bleeding, sepsis, and thromboembolic events. Information on thromboembolic events (type, date of occurrence, and localization) was gathered from medical records as well as radiology findings. The study period encompassed the entire duration of ECMO support as well as 10 days after ECMO termination, since certain diagnostic tests (computed tomography, ultrasound) may not have been performed during ECMO but after its termination. According to its localization, thrombosis was further divided into venous and arterial.

Information on hemorrhage was collected merely throughout ECMO. We adapted the ELSO bleeding definition, dividing it further into minor and major hemorrhages [25]. A major hemorrhagic event was defined as clinically overt bleeding followed by a hemoglobin reduction of at least 2 g/dL within one day (24 h) or administration of at least two red blood concentrates over the same time. A major hemorrhage further included any retroperitoneal or pulmonary bleeding and bleeding requiring surgical intervention or involving the central nervous system. Any other noticeable bleeding was defined as minor. If the bleeding event occurred repeatedly, only the date of the first event was recorded.

Death-related data (date and cause) were retrieved from the electronic documentation or postmortem examinations, if available. Based on the reported date of death, mortality in different periods was calculated.

This work was approved by the Ethics Committee of the Medical University of Innsbruck, Austria (Ethics Committee Number: 1274/2019).

### 2.4. Statistical Analyses

SPSS was used for statistical analyses (Version 28.0., IBM Corp.: Armonk, NY, USA, 2021). A two-sided *p*-value of less than 0.05 was considered significant. Based on the data normality and variable type, we present results as median with minimum and maximum, mean with standard deviation, or frequency with percent. The independent samples *t*-test was used for parametric data and Mann–Whitney U test for numeric/ordinal data (non-normal distribution). Fisher’s exact and Chi-square tests were employed to analyze nominal data. In the univariate Cox regression analyses, we estimated the effect of each potential risk factor on mortality, and all significant variables were thereafter assessed in the multivariate model. The significance level for the multivariate model was set to 0.05. We repeated multivariate models including different parameters to explore the association of mortality and different blood biomarkers with mortality throughout the ECMO.

## 3. Results

### 3.1. Patient Characteristics

In total, 188 patients were included in the final analysis. The ECMO indication was cardiogenic shock due to myocardial infarction. The median SOFA and SAPS III scores were 11 (1–21) and 66 (31–104), respectively. Survivors had lower SAPS III scores compared to deceased patients (Table 1). Almost 40% (37,8%, 71/188) of patients were resuscitated before or during ECMO implantation, with 21% (40/188) resulting in mortality.

The average ICU length of stay was 20 (1–79) days, and 53.2% (100/188) of patients survived the ICU and were discharged from the hospital (Table 1). Overall, 41 patients (22%, 41/188) died during ECMO support and an additional 39 during the ICU stay. The main cause of death was cardiac (21%, 40/188), followed by multiple organ dysfunction syndrome (11%, 20/188) and sepsis (6%, 12/188).

The median duration of support was 6 (1–22) days; in 72% (135/188) of patients, ECMO was required for less than seven days (Table 2). Patients were anticoagulated mainly with UFH (139/188, 74%) and argatroban (24/188, 13%). Due to severe hemorrhage, 10% (18/188) of patients were not anticoagulated at all.

### 3.2. Laboratory Parameters during ECMO

Nineteen blood parameters were analyzed within the entire observational period. C-reactive protein (CRP) levels rose analogously fast in compared groups up to the sixth day of support, when CRP levels re-increased in decedents while remaining stable in survivors. After termination of ECMO, CRP levels were lower in survivors on the third and especially the tenth day (Figure 1).

Procalcitonin reached its peak on day three in both groups, with the maximum in deceased patients being more than twice as high compared to survivors (median 13 mcg/L vs. 27 mcg/L in survivors and non-survivors, respectively (Figure 2)). Thereafter, PCT decreased markedly in both groups until the sixth day. Thereafter, PCT decreased in both groups, remaining higher in deceased patients. Following ECMO termination, PCT levels were higher in deceased patients, especially on day 10 (median 1.4 mcg/L in survivors vs. 21 mcg/L in deceased patients).

Within the first day, fibrinogen levels reduced by about one-third and started increasing from the second day. However, no major differences were observed between survivors and the deceased, both during and after the termination of ECMO support (Figure 3).

Finally, platelets decreased to more than half of their initial levels, reaching the nadir on the sixth day (Figure 4). The platelet count was always slightly higher in survivors compared to deceased patients. After day 6, the platelet count started to increase in both groups. However, the platelet counts never reached baseline levels during ECMO support. Following the termination of ECMO support, the platelet counts markedly increased in both groups, with survivors having a higher platelet count compared to deceased patients.

### 3.3. Adverse Events

Hemorrhage was the most frequent complication (86/188, 46%), followed by thrombosis (50/188, 27%) and sepsis (34/188, 18%) (Table 2). Overall, 43% (80/188) did not survive to hospital discharge, with cardiac failure (40/188, 45%) and multiple organ dysfunction syndrome (20/188, 23%) as the main causes of death (Table 1).

We observed hemorrhage more often in deceased patients (45/86, 52%, *p* = 0.018), including a higher portion of major bleeding (25/188, 32%, *p* = 0.007). The overall incidence of thrombosis was similar between the compared groups. However, arterial thrombosis occurred more often in deceased patients compared with survivors (25% vs. 16%, *p* = 0.147). Finally, deceased patients experienced sepsis more often compared with survivors, even though this did not reach statistical significance (19.3% vs. 17%, *p* = 0.708).

The multivariate Cox regression model showed that higher levels of PCT on the second, third, fourth, and fifth days were associated with increased hazard ratios for in-hospital mortality. In cases of adverse events, bleeding events during ECMO showed a trend toward increased mortality (Table 3). Univariate analyses are presented in Appendix A and additional multivariate models in Appendix A.

## 4. Discussion

In this study, we investigated trends of blood biomarkers commonly used during va-ECMO support due to MI. In our study population, non-survivors had significantly higher blood levels of PCT throughout the entire observation period. Regarding adverse events, ECMO support was frequently complicated by bleeding and thromboembolic events, with an overall ICU mortality rate of 43%. Finally, higher blood PCT levels over the course of ECMO and hemorrhage during or shortly after support have been shown to be associated with increased hazard ratios for in-hospital mortality.

We observed a rather low mortality rate compared to other studies published. For instance, a recent meta-analysis on mortality in cardiogenic shock due to acute MI in va-ECMO patients reported a one-year survival rate of 23% to 36% [2]. These discrepancies could be explained by a diverse patient population with distinctive risk factors or even various definitions of cardiogenic shock. Moreover, we excluded patients who had ECMO support for less than 24 h, which may potentially bias the low mortality rate. However, the majority of studies reporting on predictors for mortality or adverse events during ECMO support exclude this patient population, as the identification of risk factors in support shorter than one day would be very complex [11].

Decedents were sicker than survivors before the commencement of ECMO (as measured with the SAPS III score) and experienced hemorrhage more often. Clearly, hemorrhage during ECMO occurs more often in decedents and is a well-established risk factor in the current literature [10,11]. In a recent meta-analysis of studies dealing with ECMO support in cardiogenic shock, renal failure with the need for renal replacement therapy followed by hemorrhage were the most often reported adverse events [11]. Moreover, therapeutic anticoagulation is a subject of discussion, and studies on anticoagulation-free ECMO support are occurring [26,27,28,29,30,31]. Hemorrhage is often identified as a risk factor for mortality, while thromboembolic events did not show increased hazard ratios for mortality [31]. However, the evidence on the health-related quality of life after ECMO support is still limited, reporting on an overall positive outcome [32,33,34]. However, there are no studies reporting on the health-related quality of life focusing on patients surviving ECMO-associated adverse events, and it may be that the level of dependency after ECMO-associated adverse events is high with poor outcomes. Finally, the majority of studies investigating ECMO support evaluated only the survival rate to hospital discharge or transfer, missing the component of quality of life.

In our study, PCT has been shown to be associated with higher in-hospital mortality in patients with cardiogenic shock. With PCT values being higher in deceased patients during ECMO support, this time course is comparable with recent data [35,36,37]. A study published in 2006 has already demonstrated that increased PCT levels are associated with increased mortality in critically ill patients. Similar to our study, the investigators did not find an association between either CRP levels or WBC count and increased mortality [38]. Moreover, it is well established that acute MI induces an inflammatory response [39]. Indeed, it has been shown that PCT levels correlate with MI severity. Uncomplicated MI has not been shown to be associated with increased PCT levels, but as the severity increased and MI was complicated with pulmonary edema or cardiogenic shock, PCT levels differed markedly from those of patients with uncomplicated MI [40]. Higher PCT levels in patients undergoing ECMO support do not necessarily imply an infectious etiology. Indeed, ECMO support has been associated with an inflammatory response being triggered (or at least favored) by the continuous exposure of blood to the artificial surface of the ECMO circuit and the surgical trauma at the cannulation site [30,41]. A small retrospective study including 38 patients investigated the role of PCT in predicting infection and survival in patients with cardiogenic shock undergoing ECMO support. The authors found that PCT was not useful in predicting the occurrence of new nosocomial infections during ECMO support; however, increased PCT levels within the first week of ECMO support were associated with a significantly higher mortality rate [42].

Clearly, these patients represent a challenge for every intensivist, as both the increased incidence of bacterial infections as well as the inflammatory response mounted by the underlying disease can contribute to increased PCT levels, and the distinction may be rather complex.

In a recently published review of va-ECMO support in the management of cardiogenic shock, the authors describe the existence of only two published randomized controlled trials to date examining va-ECMO support in patients with cardiogenic shock [43]. In the ECMO-CS trial, patients were randomized to va-ECMO support or conservative management, with the possibility of va-ECMO support in cases of hemodynamic deterioration. No significant differences could be observed in the incidence of adverse events or all-cause mortality between the two groups [44]. In the EUROSHOCK trial, patients were randomized to va-ECMO vs. standard therapy. The trial included 35 patients in total. Patients who underwent va-ECMO support had a significantly better outcome, measured as all-cause mortality at thirty days and one year. However, these patients also had a markedly increased rate of adverse events, specifically bleeding events and vascular complications [45].

A recently published randomized controlled trial sought to investigate whether ECMO support is beneficial in patients with MI complicated by cardiogenic shock [46]. Four hundred and seventeen patients were included in the final analyses. The authors could not show a survival benefit at 30 days in the ECMO group compared to the control group. Bleeding occurred significantly more often in the ECMO group, which is consistent with the current literature and may have at least partly contributed to the fact that ECMO support was not superior to the standard of care. In their study, the investigators did not focus on blood biomarkers, which was a focus in our study. Even though, according to the recent randomized controlled trial, ECMO support does not appear to confer a survival benefit in this patient group, increased PCT levels seem to portend a worse clinical outcome. However, whether PCT is a marker of the increased risk for adverse events or a marker of inflammation (potentially a modifiable factor) remains to be elucidated.

### Limitations

Certain limitations should be kept in mind. As this was a retrospective analysis, all the limitations pertinent to retrospective studies, including selection bias, also apply. Moreover, differentiating whether the presence of an infection or the actual illness causes a change in the levels of biomarkers, especially PCT levels, in critically ill patients is challenging. The levels of fibrinogen, platelets, hemoglobin, and antithrombin were not evaluated in greater detail, as the substitution of blood and coagulation products could have influenced the measured levels. Even though this is a large study addressing patients with acute MI complicated by cardiogenic shock undergoing ECMO therapy, it is not possible to completely exclude the potential effect of missing data. This may hold especially true, as we were not able to analyze other parameters that may have an impact on the patient outcome (kidney failure, the need for continuous renal replacement therapy, all potential comorbidities, thrombolysis in myocardial infarction III (TIMI III) flow, localization of myocardial infarction, antiplatelet therapy received, troponin level, etc.) [47,48]. However, we sought to objectify the degree of sickness in our study population using the SOFA and SAPS III scores. Moreover, it may be quite challenging to discriminate potential complications of the underlying illness from ECMO-related adverse events. Finally, we analyzed ECMO outcomes over a period of more than ten years. As therapy for MI has evolved substantially during this time span, patients have likely been included in this analysis who potentially received different therapies. However, sensitivity analysis did not show significant differences in outcomes over the years.

## 5. Conclusions

We provide a study focusing on the time course of the most common inflammatory blood biomarkers and their role in the outcome of ECMO patients with MI-associated cardiogenic shock. Non-survivors had increased blood levels of PCT starting from day one of ECMO support, which remained increased throughout the observation period. The multivariate model identified PCT as a blood biomarker with increased hazard ratios for mortality. Regarding adverse events, ECMO support was frequently complicated by bleeding and thromboembolic events, with an overall ICU mortality rate of 43%. Future studies are needed to show the role of PCT in patients receiving ECMO support.

## Figures and Tables

**Figure 1 diagnostics-13-03683-f001:**
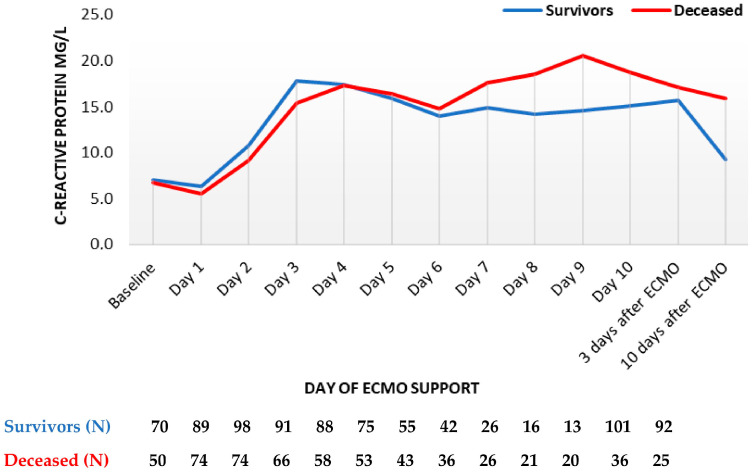
C-reactive protein trend throughout support (mean values). ECMO, extracorporeal membrane oxygenation. Data are presented as mean.

**Figure 2 diagnostics-13-03683-f002:**
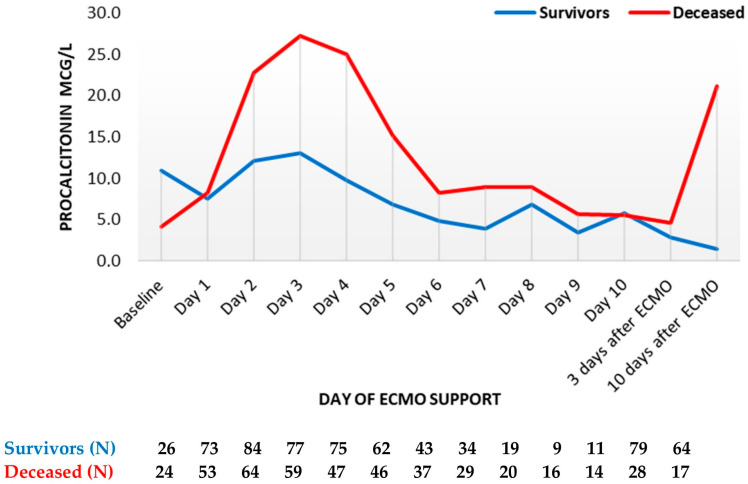
Procalcitonin trend throughout support (mean values). ECMO, extracorporeal membrane oxygenation. Data are presented as mean.

**Figure 3 diagnostics-13-03683-f003:**
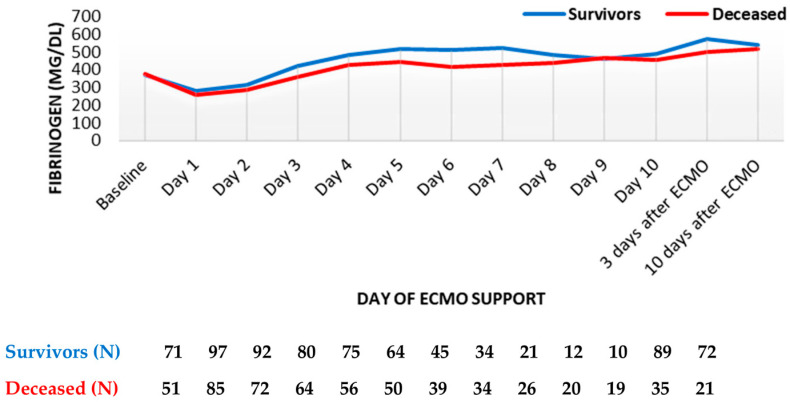
Fibrinogen trend throughout support (mean values). ECMO, extracorporeal membrane oxygenation. Data are presented as mean.

**Figure 4 diagnostics-13-03683-f004:**
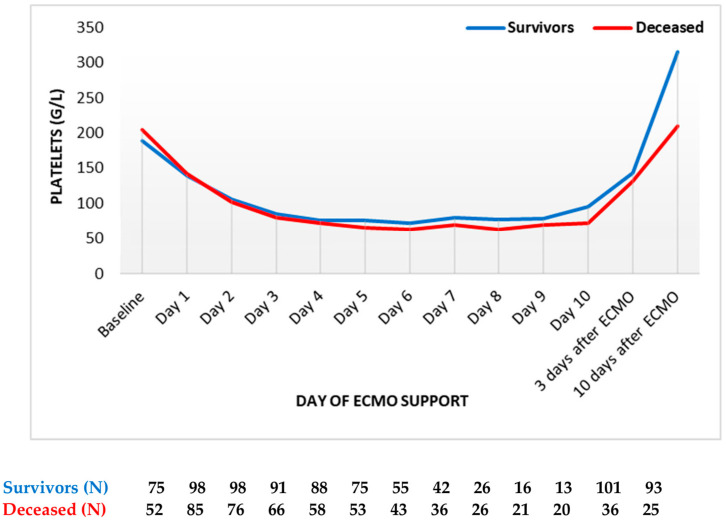
Platelets count trend throughout support (mean values). ECMO: extracorporeal membrane oxygenation. Data are presented as mean.

**Table 1 diagnostics-13-03683-t001:** Clinical characteristics of included population (*n* = 188).

Patient Characteristics	All Patients (*n* = 188)	Survivors (*n* = 100)	Deceased (*n* = 88)	*p*-Value
Age (years)	62.8 ±10.5	62.9 ±10.4	62.7 ±10.7	0.905
Male sex	147 (78.2)	81 (81.0)	66 (75.0)	0.377
Body mass index (kg/m^2^)	26.7 ±4.0	26.7 ±4.2	26.7 ±3.8	0.968
SAPS III score	66 (31–104)	62 (31–99)	70 (33–104)	<0.001
SOFA score	11 (1–21)	11 (3–21)	11 (1–19)	0.422
CPR before ECMO initiation	71 (37.8)	31 (31.0)	40 (45.5)	0.050
Length of ICU stay (days)	20 (1–79)	25 (4–79)	14 (1–74)	<0.001
Mortality				
Death during ECMO support	41 (21.8)	-	41 (46.6)	
Death during ICU	80 (42.6)	-	80 (90.9)	
Death within 60 days	84 (44.7)	-	84 (95.5)	
Death within 90 days	88 (46.8)	-	88 (100.0)	
Cause of death				
Cardiac	40 (21.3)	-	40 (45.4)	
MODS	20 (10.6)	-	20 (22.7)	
Sepsis	12 (6.4)	-	12 (13.6)	
Brain death	10 (5.3)	-	10 (11.4)	
Unknown cause	6 (3.2)	-	6 (6.9)	

Abbreviations: SAPS, simplified acute physiology III score; SOFA, sequential organ failure assessment score; ECMO, extracorporeal membrane oxygenation; MODS, multiple organ dysfunction syndrome; ICU, intensive care unit; CPR, cardiopulmonary resuscitation.

**Table 2 diagnostics-13-03683-t002:** ECMO-related characteristics and outcomes (*n* = 188).

Clinical Characteristics	All Patients (*n* = 188)	Survivors (*n* = 100)	Deceased (*n* = 88)	*p*-Value
ECMO-related clinical course				
ECMO support duration (days)	6.3 (1–22)	6.2 (2–14)	6.5 (1–22)	0.625
ECMO support duration <7 days	135 (71.8)	74 (74.0)	61 (69.3)	0.518
Anticoagulation during ECMO support				
Unfractionated heparin	139 (73.9)	78 (78.0)	61 (69.3)	0.124
Argatroban	24 (12.8)	13 (13.0)	11 (12.5)	
Epoprostenol	1 (0.5)	0 (0.0)	1 (1.1)	
None	18 (9.6)	5 (5.0)	13 (14.8)	
Complications				
Hemorrhage	86 (45.7)	38 (38.0)	48 (54.5)	0.028
Major hemorrhage	35 (22.3)	10 (12.8)	25 (31.6)	0.007
Thromboembolic events	50 (26.6)	25 (25.0)	25 (28.4)	0.623
Thrombosis venous	26 (13.8)	16 (16.0)	10 (11.4)	0.403
Thrombosis arterial	38 (20.2)	16 (16.0)	22 (25.0)	0.147
Sepsis	34 (18.1)	17 (17.0)	17 (19.3)	0.708

Abbreviations: ECMO, extracorporeal membrane oxygenation.

**Table 3 diagnostics-13-03683-t003:** Multivariate analysis: identification of risk factors for mortality (*n* = 188).

Variable	B-Coefficient	*p*-Value	HR *	95% CI	
				Lower	Upper
Procalcitonin on day two	0.008	0.010	1.008	1.002	1.014
Bleeding event during or after ECMO	0.452	0.067	1.571	0.968	2.550
Resuscitation before ECMO initiation	0.211	0.410	1.235	0.747	2.042

* For every increase in one unit of measurement, hazard ratio increased by 1%. Abbreviations: CI, confidence intervals; HR, hazard ratio; ECMO, extracorporeal membrane oxygenation.

## Data Availability

The datasets used and analyzed during the current study are made available from the corresponding author on reasonable request.

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
