# Peer review of "ECMO in Myocardial Infarction-Associated Cardiogenic Shock: Blood Biomarkers as Predictors of Mortality"

_diagnostics, 2023, doi:10.3390/diagnostics13243683_

Round 1
Reviewer 1 Report
Comments and Suggestions for Authors
The authors have analysed a large cohort of patients who used Veno-arterial extracorporeal membrane oxygenation (va-ECMO). The data are well presented and are useful for other groups who may use ECMO in their daily medical practice.
All the figure legends shall be detailed, such as the data are Means or median, error bars and number of values etc.
If possible, the error bars shall be included in the figures
Comments on the Quality of English Language
Good. need a proof reading
Reviewer 2 Report
Comments and Suggestions for Authors
I reviewed with interest the manuscript "ECMO in Myocardial Infarction-Associated Cardiogenic Shock: Blood Biomarkers as Predictors of Mortality" by Senoner et al. The authors in their article studied the association of commonly used laboratory blood parameters with patient outcomes, especially focusing on the time course during the ECMO support. They were able to show that patients who died had elevated blood levels of procalcitonin starting on the first day of ECMO support, which remained elevated throughout the follow-up period. In the multivariate model, procalcitonin level on day 2 of ECMO was the only risk marker for death. However, during the review I had questions and comments to which I would like to receive answers from the authors.
1. It is important that the article does not contain clinical characteristics of the patients included in the study. Judging by the authors’ logic, mortality in such patients depends only on ECMO related characteristics. However, in a meta-analysis, failure to achieve TIMI III flow and left main artery were highly associated with mortality in ECMO in Myocardial Infarction-Associated Cardiogenic Shock. Older age, BMI greater than 25 kg/m2, renal dysfunction, increased lactate, decreased prothrombin activity, VA-ECMO implantation after PCI, and the presence of non-shock rhythm were factors associated with increased mortality for which there was low to moderate confidence in the estimates (Sohail S et al, 2022, ref. 7 from the article). None of these indicators were studied in this article, which reduces the scientific value of the results obtained.
2. I think it is also necessary to take into account the following indicators characterizing the course of myocardial infarction: localization of myocardial infarction, troponin level, invasive coronary angiography data, PCI, antiplatelet therapy received. Also, these same indicators needed to be included in multiple logistic regression models.
3. Authors should add consideration of recent publications on the topic of the manuscript, such as the review by Koziol et al (2023, ref. 1, see below).
4. In graphs 1-4 there is no statistical data on the reliability of differences between groups. Also, this information is not in the text of the manuscript.
5. The authors have already published several studies on this topic. Therefore, it is necessary to justify in more detail exactly what new results are presented in this article. For example, graphs 1-4 are present in the authors’ article published in 2022 (ref. 2, see below).
References:
1. Koziol KJ, Isath A, Rao S, Gregory V, Ohira S, Van Diepen S, Lorusso R, Krittanawong C. Extracorporeal Membrane Oxygenation (VA-ECMO) in Management of Cardiogenic Shock. J Clin Med. 2023 Aug 26;12(17):5576. doi: 10.3390/jcm12175576.
2. Rajsic S, Breitkopf R, Oezpeker UC, Treml B. ECMO in Cardiogenic Shock: Time Course of Blood Biomarkers and Associated Mortality. Diagnostics (Basel). 2022 Nov 26;12(12):2963. doi: 10.3390/diagnostics12122963.
Comments on the Quality of English LanguageNo comments
Round 2
Reviewer 2 Report
Comments and Suggestions for Authors
The authors answered my questions in detail and also made some clarifications to the text of the manuscript.
Comments on the Quality of English LanguageNo comments